# SVM Regression to Assess Meat Characteristics of Bísaro Pig Loins Using NIRS Methodology

**DOI:** 10.3390/foods12030470

**Published:** 2023-01-19

**Authors:** Lia Vasconcelos, Luís G. Dias, Ana Leite, Iasmin Ferreira, Etelvina Pereira, Severiano Silva, Sandra Rodrigues, Alfredo Teixeira

**Affiliations:** 1Mountain Reserach Center (CIMO), Polytechnic Institut of Bragança, Campus de Santa Apolónia, 5300-253 Bragança, Portugal; 2Laboratory for Sustainability and Technology in Mountain Regions, Polytechnic Institut of Bragança, Campus de Santa Apolónia, 5300-253 Bragança, Portugal; 3Veterinary and Animal Research Centre (CECAV), Associate Laboratory of Animal and Veterinary Science (AL4AnimalS), University of Trás-os-Montes e Alto Douro, 5000-801 Vila Real, Portugal

**Keywords:** NIR, SVM model, meat quality, Bísaro pig, *Longissimus thoracis et lumborum*

## Abstract

This study evaluates the ability of the near infrared reflectance spectroscopy (NIRS) to estimate the aW, protein, moisture, ash, fat, collagen, texture, pigments, and WHC in the *Longissimus thoracis et lumborum* (LTL) of Bísaro pig. Samples (*n* = 40) of the LTL muscle were minced and scanned in an FT-NIR MasterTM N500 (BÜCHI) over a NIR spectral range of 4000–10,000 cm^−1^ with a resolution of 4 cm^−1^. The PLS and SVM regression models were developed using the spectra’s math treatment, DV1, DV2, MSC, SNV, and SMT (*n* = 40). PLS models showed acceptable fits (estimation models with RMSE ≤ 0.5% and R^2^ ≥ 0.95) except for the RT variable (RMSE of 0.891% and R^2^ of 0.748). The SVM models presented better overall prediction results than those obtained by PLS, where only the variables pigments and WHC presented estimation models (respectively: RMSE of 0.069 and 0.472%; R^2^ of 0.993 and 0.996; slope of 0.985 ± 0.006 and 0.925 ± 0.006). The results showed NIRs capacity to predict the meat quality traits of Bísaro pig breed in order to guarantee its characterization.

## 1. Introduction

Near Infrared Spectroscopy (NIR) has emerged as an efficient, fast, and non-invasive instrument used for analysis, classification, monitoring, and predicting qualitative and quantitative information. The well-described advantages of NIR suit the food processing industry in terms of operating speed and possible implementation of in-line, on-line, or at-line process monitoring. It could also meet consumer expectations in terms of product quality and safety assurance. Coupled with advanced chemometric tools, these high throughput yet cost-effective tools have shifted the focus away from lengthy and laborious conventional methods (physicochemical, instrumental, and sensory). They require sample preparation procedures and are therefore not applicable to the fast-paced industrial meat sector, which has become unsuitable for real-time analysis and generates hazardous waste [1,2,3,4,5,6].

The NIR is a widely used methodology for some chemical analysis in foods because it provides complete information about the molecular bonds and chemical constituents in a sample, being a convenient tool not only to characterize foods but also to evaluate the quality and control of the processes [4,7]. The NIR spectra include broad bands from overlapping absorption profiles, corresponding mainly to overtones and combinations of vibrational modes involving chemical bonds. These bonds in the NIR spectra show important information of a sample’s composition [7]. Its calibration requires a multivariate mathematical model able to relate the spectral absorption of the near infrared region with analytical data obtained from a reference method for each parameter of interest. A large database is essential to the development of robust and accurate NIRs predictions [8]. Besides that, NIR spectral analysis remains prone to ambiguities, which translate to convoluted spectral changes [9]. Hence, pre-treatment techniques, mainly including multiplicative scatter correction (MSC), standard normal variate (SNV), smoothing (SMT), baseline removal, and first derivate (DV1) and second derivative (DV2), are used to reduce and correct possible interferences related to scattering, baseline shift, path-length variation, and overlapping bands.

Multivariate statistical techniques, such as principal component analysis (PCA), soft independent modelling of class analogy (SIMCA), projections algorithm (SPA), and k-nearest neighbors’ algorithm (KNN) are the most used in the qualitative explorative analysis. For the sample’s identification, characterization, and discrimination (supervised qualitative analysis), linear discriminant analysis (LDA), partial least square discriminant analysis (PLS-DA), and support vector machine (SVM) are some examples of the multivariate techniques applied in data modeling. Multiple linear regression (MLR), principal component regression (PCR), partial least square regression (PLSR), support vector machine (SVM), and artificial neural network (ANN) are commonly used for modeling purposes and are also used for quantitative purposes [10,11].

Several studies have confirmed that NIR has an unlimited potential to assess meat quality, being suitable for all meat species in large-scale quality evaluation [4]. Among many others, NIR can rapidly analyze meat color, pH, and tenderness in intact fresh beef [12]; moisture, protein, and fat content of beef burger [13]; minced lamb tenderness and different types of sheep meat [14,15]; protein, moisture, connective tissue, and ash content in goat minced loin [16]; and springiness, moisture, ash, protein, lipids, pH, and color of different minced parts of fresh chicken [17,18]. In pork meat, NIRs was used to determine moisture content, protein, and intramuscular fat in different sets of intact and minced samples [1,19,20,21,22]. It was also used to analyze the drip loss, color, and pH of commercial intact pork loins [23,24]; intact loins’ water holding capacity (WHC) [25]; color (L*, a*, and b*), myoglobin, centrifuge force water loss, and texture, as Shear Force (SF) and texture profile analysis [26].

The meat industry could benefit from a system that can predict variations in single attributes or even classify samples according to quality features. Regarding pork, its quality is based on various attributes: carcass commercial value, meat organoleptic, and nutritional and technological properties (i.e., suitability for processing and storage) [27]. More recently, there has been a large increased interest in the non-compositional aspects of meat-related quality, as well as the intrinsic characteristics of the animals (species, breeds), the geographical origins, the food received, the productive management, or the post-mortem strategies [4]. This produces a significant amount of fresh pork and pig products that are marketed or sold under official quality seals.

The Bísaro pig is an autochthonous breed in the north of Portugal whose meat and meat products are of a recognized high quality. Its producers have shown a growing interest in the production of premium meat products, having two types of carcasses with protected designation of origin (PDO) meat products [28].

To obtain high quality meat products from the Bísaro pig with greater added value, which is particularly relevant to ensure the Protected Designation of Origin (PDO) and Protected Geographical Indication (PGI) origin brands, it is essential to have quick analytical tools to control the quality of the meat processing, which is a necessity to disseminate this breed for the consumers.

The aim of the present study was to evaluate the accuracy of the PLS and SVM models to determine and estimate some important meat quality characteristics in the minced samples of loins belonging to Bísaro pigs fattened in an open-air system using the near infrared region.

## 2. Materials and Methods

This study is a part of a project (BISIPORC project, financed by the PRODER program, measure 4.1 Cooperation for Innovation) between the National Association of Bísaro Pork Breeders, a research center (Carcass and Meat Quality Laboratory at the School of Agriculture of the Polytechnic Institut of Bragança), and a meat manufacturing industry (Bísaro Salsicharia Tradicional^®^) to develop and add value to the animals reared in the extensive system.

### 2.1. Animals and Slaughter Procedure

A group of 40 animals were fed on a Bísaro pork meat farm for 90 days until they reached an average body weight of 100 kg. When they reached the desired body weight, pigs were slaughtered in the Municipal Slaughterhouse of Bragança, Portugal. The procedure for slaughtering and cutting the carcass was previously described by Álvarez-Rodríguez and Teixeira [29]. A total of 40 carcasses were evaluated. All animals were cared for and killed in compliance with the welfare regulations and respecting EU Council Regulation (EC) No. 1099/2009 [30].

After slaughter, the carcass weight was recorded and then placed in a cooling chamber. After cooling at 4 °C for 24 h, the cold carcass weight was recorded.

### 2.2. Physicochemical Analysis and Chemical Composition

The carcasses were carefully halved, and the left side was weighed and recorded. The carcasses were carried to the Carcass and Meat Quality Laboratory at the School of Agriculture of the Polytechnic Institut of Bragança (Portugal) for carcass evaluation and meat analysis. *Longissimus thoracis et lumborum* (LTL) muscle samples were obtained between the 7th and 12th rib through dissection of the carcass for physicochemical analysis. Part of each muscle sample was ground using a power mill Buchi Mixer B-400 (BÜCHI, Labortechnik AG, Postfach, Flawil, Switzerland), around 5 to 10 s, for obtaining homogeneous paste (weight around 100 g). The rest of the LTL sample was used for other analysis such as WHC and SF. The water activity (aW) analyses were carried out according to AOAC [31] using a probe HigroPalmAw1 Rotronic 8303 (Bassersdorf, Switzerland). The determination of moisture was performed according to NP-ISO-1614/2002 [32]; briefly, approximately 3 g of the sample were added to 5 mL of ethanol. After that, the samples were oven-dried (Raypa DO150, Barcelona, Spain) for 24 h at 103 ± 2 °C and the lost mass of water was measured. For ash content, the samples were incinerated at 550 ± 25 °C during 5–6 h in muffle furnace (Vulcan BOX Furnace Model 3-550, Yucaipa, CA, USA) and we measured the mass of ash obtained according to NP-ISO-1615/2002 [33]. The collagen content via hydroxyproline determination following NP 1987/2002 [34] and protein content were analyzed using the Kjeldahl method in accordance to NP-ISO-1612/2002 [35]. WHC was assessed according to the Honikel procedure, samples of LTL muscle (100–120 g) were cooked inside plastic bags in a 70 °C water bath until reaching 70 °C, measured in the muscle center, and the samples were weighed after 30 min of rest [36]. SF was evaluated in raw (RT) and cooked (CT) samples using an INSTRON 5543J-3177 equipped with a Warner–Bratzler device. Approximately 8 muscle sub-samples (1 cm^2^ cross-section) were taken from each muscle for SF evaluation. The measurement was recorded as the average yield force in kilograms (Kgf), required to perpendicularly shear to the direction of the fibers. All procedures were carried out at room temperature [37]. Haem pigments were obtained using the reflectance of the exposed surface by spectroscopy with a Spectronic Unicam 20 Genesys (SPECTRONIC 20 GENESYS, Thermofisher Scientific, Austin, TX, USA) at 512 nm and the results are expressed in mg myoglobin/g fresh muscle. The method is based on the muscle pigment content procedure defined by Hornsey [38]. The intramuscular fat (total lipidic content) was extracted from 25 g of meat sample according to the Folch procedure [39]. All analyses were performed in triplicate.

### 2.3. Sample Set and NIRS Analysis

Samples of LTL Bísaro pig muscle (*n* = 40) were minced, as referred to in Section 2.2, and placed in petri dishes (diameter around 9 cm). A FT-NIR MasterTM N500 (BÜCHI) prepared with a 360^°^ rotation system was used. This instrument operates between 4000 and 10,000 cm^−1^ spectral range with a resolution of 4 cm^−1^. NIRCal BÜCHI software, version 5.5, was used to save spectra data into an Excel^TM^ file. Three spectra per sample were measured and used for the development of the calibration equations.

### 2.4. Statistical Analysis

All data treatment was carried out using the open-source software R (application Intel, R.app GUI 1.78, © R Foundation for Statistical Computing, 2021) e RStudio (application Intel, RStudio 2022.02.3 Build 492, © 2009–2022 RStudio, PBC).

The independent data corresponds to the NIRs spectrum obtained from the LTL samples analyzed.

Several data spectrum treatments (processing) were considered [40]: SMT; DV1, and DV2 derivatives; normalization (NORM) to unit area and correction of spectral baseline as SNV; MSC; and asymmetric least squares (ALS). All these treatments, together with the smoothing processing; and, also, the combinations of the normalization and correction of spectral baseline with DV1.

Considering that a spectrum represents a set of correlated data, whose number of data exceeds the number of analyzed samples, a selection of variables was performed to reduce to 10% of its information. For this, points were selected with a wave number interval of 20 cm^−1^, allowing to reduce the spectrum of 1501 points to 151 points. The 11 processing methods with this variable reduction coupled with the SVM-Poly allowed us to obtain the estimation and predictive models of the dependent variables.

To describe the data variability within all dependent variables obtained from the 40 samples, the minimum, maximum, and average with the standard deviation values were used. In addition, the correlation matrix was applied to verify the correlation between all the combinations of the two dependent variables [41].

Dependent data were divided into 2 groups: train group, with 32 samples (80%); test group, with 8 samples. The procedure used was carried out by the algorithm Kennard-stone, a uniform mapping algorithm from the prospect package [42]. This algorithm is based on the principal components of the independent variables, and it allowed us to ensure that the same samples were used to test the model’s predictive ability (ensuring the independence of the samples from those of the training data subset). The train data subset was used for each model’s training and the test data subset for external validation, to confirm the model’s performance in predicting new samples. For this, the R’s caret package [43] was applied to each sample’s dependent variables. In addition, a cross-validation with 8 folds and 10 repetitions [43] was applied to the model’s training as an internal validation procedure (it implies the evaluation of the predictive performance of 80 different models), allowing us to use the limited train data to estimate the model’s performance, helping to avoid overfitting.

The support vector machine regression (SVMR) technique, a supervised learning model for regression analysis, relies on kernel functions to construct the models. SVMR does not depend on the distributions of the underlying dependent and independent variables. The commonly used kernel functions are: (a) linear, (b) radial basis, and (c) polynomial. The selection of the appropriate kernel function depends on the quantitative data, and it requires optimization techniques for the best model’s selection. The first kernel used was the linear, which gave poor prediction results, therefore non-linear ones were used. Since a non-linear relation between the variables of interest was expected, the radial basis function (RBF) kernel (the kernel function transforms the data from non-linear space to linear space) and the polynomial function (PF) kernel were used. However, these results were not presented in this work because the SVMR-Poly models showed better results for both estimation and prediction. With SVMR using the PF kernel, one more parameter was evaluated—the degree of the polynomial function [44]—making the decision boundary more flexible. Thus, a grid search for the optimization of C, scale, and degree was also performed to choose their optimal values.

The predictor variables were centered and scaled. The average of the root mean square error (RMSE) and the mean absolute error (MAE) were used as the predictive evaluation criteria. To visualize and evaluate the NIR capability to quantify each meat quality parameter, a simple linear regression model was established between the predicted model and real values for train and test data groups. The results were considered satisfactory if the linear regression parameters were close to the theoretical values [45,46]: “zero” (0) for relative standard error (RSE) and intercept; “one” (1) for slope and the determination coefficient. In addition, the confidence interval at 95% of the slope and intercept were used to statistically infer if they could be equal to the theoretic values of “one” and “zero”, respectively.

## 3. Results and Discussion

### 3.1. Physicochemical Analysis and Chemical Composition

To achieve the simultaneous detection of multiple parameters for meat quality, in this study, the aW, moisture, ash, collagen, WHC, pigments, CT, RT, fat, and protein of 40 different samples of LTL muscle were predicted as the first quality calibration indices to Bísaro pig. Table 1 presents a summary of the results obtained for these variables.

These 10 parameters were related to each other through a pairwise correlation matrix. It was found that, in general, the absolute values of Pearson’s correlation coefficients were equal or lower than 0.59, except for the relationship between fat and moisture, which presented a correlation coefficient of −0.84. The data exhibited relatively large ranges and high standard deviations, showing acceptable sample variability. Similar ranges were found in published articles related to different pork quality assessments using NIR analysis. A study conducted by Cáceres-Nevado et al. [21], with 277 intact and minced Iberian pig loins (located 10 cm away from the head) employing the PLS model, for training and calibration set with full spectra range (831–2502 nm), reached minimum and maximum values of around 1.66–15.20% for fat, 64.89–74.45% for moisture, and 17.80–23.87% for protein, respectively. In another work, the same authors [6], using similar conditions (524 intact and minced Iberian *Longissimus dorsi* samples located between the 13th and 14th dorsal vertebrae, applied a modified partial least square (MPLS) regression, which allowed to get, in the calibration and validation set, results with range of 1.16–12.90% for intramuscular fat, 66.20–74.70% for moisture, and 19.06–24.00% for protein content. The results of the present work were quite similar to those reported by these authors. Other examples are: the study by Zamora-Rojas et al. [1], on 348 ground Iberian pig samples (*gluteus medius*, *masseter*, *Longissimus dorsi,* and *spinalis dorsi*) with the same MPLS model, obtained for training, calibration test, and recalibration set with full spectra (400–2500 nm), range values of 2.30–18.30% for fat, 63.00–75.30% for moisture, and 16.90–24.70% for protein; Barbin et al. [22] studied 120 intact and minced Ireland pig muscles (*Longissimus dorsi*, *semimembranosus*, *semitendinosus,* and *biceps femoris*) with PLSR models and obtained range values for spectral information (897–1752 nm spectra range) of *Longissimus dorsi* of 0.30–6.27% for fat, 69.12–75.08% for moisture, and 22.70–25.23% for protein; Fernandez-Barroso et al. [26] studied 287 intact and minced Iberian pig samples of *Longissimus thoracis et lumborum* trough PLS model and obtained range values for myoglobin content of 1.04–2.64 mg myoglobin/g fresh muscle. However, the study by Balage et al. [2], with 134 intact *Longissimus dorsi* samples from Brazilian pigs (taken between the 9th and 11th ribs) with PLS calibration models (spectrum range from 400 to 1395 nm), reported low NIR predictability for intramuscular fat values of 0.022–0.712%, which were much lower than our results for the same parameter.

All these above results agree, in general, with those found in this work. However, it should be noted that it was not possible to find studies with the results of analysis by NIR for the parameters aW, ash, collagen, WHC, RT. and CT in pork. In this context, this work is innovative because it studied parameters not yet mentioned in the bibliography (considering the research carried out) and because it presents an extended study of prediction/estimation of 10 parameters.

Figure 1 shows the Boxplots of all dependent variables their data variability can be seen. It allows us to verify that the variables moisture, pigments, collagen, and CT have data gaps within their range of values due to the presence of extreme values, which were considered acceptable results. Moreover, it was not considered a variable transformation since they showed close to a normal distribution, even for the pigments variable, where the extremes at higher values gave a tail to its distribution.

### 3.2. NIR Spectra

Each sample was analyzed by NIR three times in the range of 4000 to 10,000 cm^−1^. The three spectra replicates were used in the multivariate analysis since it allowed us to include the variability associated with the heterogeneity of the samples and contribute to the adjustment of more robust models. Overall, the spectra presented absorbance signals that varied between 0.005 and 0.345.

To establish predictive models between the dependent and independent variables (mean spectra), the spectra were used with several treatments to select the most suitable one to obtain a predictive model: SMT, DV1, DV2, NORM-SMT, SNV-SMT, MSC-SMT, ALS-SMT, NORM-DV1, SNV-DV1, MSC-DV1, and ALS-DV1. Figure 2 shows the raw spectra and those treated with SMT, NORM-SMT, SNV-SMT, MSC-SMT, ALS-SMT, DV1 and DV2. The spectra had 1501 points, which exceeds the number of samples analyzed (40 LTL samples). Due to this discrepancy, the regression method normally applied would be the PLS.

However, other regression methods can be applied as the SVM, which can be robust in prediction since it can solve both linear and nonlinear multivariate calibration problems and learn in high-dimensional feature space with fewer training data [47].

In this work, each spectrum treatment was reduced to 10% of its initial points, selected with a wave number interval of 20 cm^−1^. Figure 3 presents the spectrum points selected (the black vertical lines) and used to obtain the quantitative predictive models. As can be seen, from the 1501 wave numbers, only 151 wave numbers were selected, which can be representative of the variations within the spectrum, since the spectrum consists of highly correlated data.

### 3.3. Quantitative Predictive Models

The data were divided into 2 groups: train group, with 32 samples (80%); test group, with 8 samples (20%). The SVMR technique, using the polynomial kernel function, was applied with cross-validation (CV) of 8 folds and 10 repetitions. The results showed that the Radial SVMR presented similar results to those obtained with the PLS model and, therefore, the data of this technique were not presented. For the SVMR with Poly kernel, three parameters were tuned (degree, C, and scale). For each dependent variable, several acceptable models were obtained, and the selection criterion was the lowest RMSE value in the train dataset and, if there were other similar options, the lowest RMSE value in the test dataset. The SVMR-Poly technique proved to be the most suitable for data modeling, then a comparison between the PLS and SVMR-Poly models was made to evaluate its performance in modeling meat characterization data. In Table 2, the cross-validation results for the selected PLS and SVMR-Poly models are presented, as well as the optimized parameters of the selected model. As can be seen, low values of RMSE, MAE, and high values of R^2^ were achieved for both PLS and SVMR-Poly models. The variability associated with the average results of the 8 models tested (cross-validation with 8 folds) can be explained by the low number of samples and, therefore, removing a significant part of the training data for internal validation has a greater impact. In general, the average RMSE, MAE, and R^2^ results of the PLS and SVMR-Poly models are similar, with a slight quality advantage for the latter. However, it appears that the PLS models, in general, need a high number of PCs, generally greater than 14, indicating that the models are complex. It was found that if the number of PCs is limited to 10, there is a decrease in the number of variables likely to have predictive models (data not shown). Regarding the independent variables most used in the PLS models, three spectrum treatments stand out: NORM-SMT for modeling data from ash, fat, and protein; SNV-SMT for moisture and pigments; SMT for aW and WHC.

The best SVMR-Poly model parameters were obtained by testing the models’ performance with the train data and with multiple models to choose from, which gave better prediction results from the test data. The SVM function had acceptable fitted model parameters, having degrees of 2, 3, and 5, small scales values (0.007 to 0.1), except for raw texture (20), and C values (0.5 to 1.3), as shown in Table 2.

ALS-DV1 was the selected spectra treatment in the calibration with the dependent variables’ ash, fat, protein, collagen, and CT, followed the DV2 spectra treatment for the dependent variables WHC and RT. This shows that SVMR-Poly uses spectral information differently and, as it allows a non-linear adjustment, it has greater versatility than the PLS technique.

Table 3 shows the general results of the predictive evaluation of the models obtained with the training and test data. The PLS models obtained showed acceptable fits (estimation models with RMSE ≤ 0.5 and R^2^ ≥ 0.95) between the meat characterization variables (dependent variables) and the independent variables (NIR signals acquired at different wave numbers, the spectrum), except for the RT variable, which showed a root mean square error of calibration (RMSE_C_) of 0.891% (high value) and coefficient of determination of calibration (R^2^_C_) of 0.748 (low value). For the variables aW, moisture, pigments, WHC, and RT, no prediction models were obtained since, in the test data, the linear relationship between the predicted and expected values for these variables showed coefficients of determination lower than 0.75 and/or slopes that were not significant or negative. For the variables ash, fat, protein, collagen, and CT, prediction models were obtained, showing R^2^_C_ greater than 0.8, slopes greater than 0.84, and non-significant intercepts, highlighting the best model for the protein variable, with a coefficient of determination of prediction (R^2^_P_) equal to 0.996 and slope_P_ 0.953 ± 0.021. The root mean square error of prediction (RMSE_P_) values for these predictive variables were lower than 3.2%.

These results are close to those found in a study conducted by Cáceres-Nevado et al. [21], with 277 intact and minced Iberian pig loins (located 10 cm away from the head) employing the MPLS regression, for the training and calibration set (831–2502 nm spectra range), which obtained a root mean square error of cross-validation (RMSE_CV_) of 0.29% and coefficient of determination of cross-validation (R^2^_CV_) of 0.98 for fat; RMSE_CV_ of 0.31% and R^2^_CV_ of 0.96 for moisture; while, for protein, the RMSE_CV_ values were 0.26% and the R^2^_CV_ values were 0.92. These authors also reached predictive values (the external validation set for minced loin) of RMSE_P_ of 0.31%, slope_P_ of 1.00, e R^2^_P_ of 0.98 to fat; RMSE_P_ of 0.38%, slope_P_ of 0.90, and R^2^_P_ of 0.93 to moisture; RMSE_P_ of 0.31%, slope_P_ of 0.92, and R^2^_P_ of 0.86 to protein. In the present work with the SVMR-Poly models, the RMSE_CV_ and RMSE_P_ for fat, moisture, and protein were much higher compared to their work.

With the SVMR-Poly models, the overall estimation and prediction results are better than those obtained by PLS. Only the variables pigments and WHC presented estimation models (respectively: RMSE_C_ of 0.069 and 0.472%; R^2^_C_ of 0.993, and 0.996; slope_C_ of 0.985 ± 0.006 and 0.925 ± 0.006), because in the test data, the R^2^_P_ values were low (<0.3) and the slope_P_’s were either non-significant or negative. The other variables presented acceptable linear relationships between the predicted and expected values for the train data and test data. The fitted models allowed linear relationships with R^2^ higher than 0.982, slope higher than 0.92, and intercept lower than 1.9. These results are representative of acceptable fit models that allowed making predictions on the test data. Acceptable linear relationships between predicted and expected values for the test data were achieved, having R^2^ greater than 0.94, slopes greater than 0.70, and non-significant intercepts. In prediction, the variables aW, moisture, and protein presented the best R^2^_P_ results (≥0.996) and acceptable RMSE_P_ values (varying between 0.001 and 0.294%), with the lowest value of RMSE for the variable aW. Satisfactorily, the variables ash, fat, collagen, and RT had R^2^_P_ values between 0.947 and 0.983, slope_P_’s close to 1, except for the collagen variable, which shows lower predictive performance as it has a slope_P_ of 0.702 ± 0.034. Although the CT variable has an R^2^_P_ of 0.840 in the linear relationship, its slope_P_ is close to unity (1.128 ± 0.100), unlike the collagen variable. These two variables are the ones with the lowest prediction performance, but they are considered possible to improve with the insertion of a greater number of samples in the database.

The predictive performance for all the best SVMR-Poly models obtained (referred to in Table 3) can be visualized in Figure 4. Globally, the results of the present study revealed that it is not possible to obtain a simultaneous prediction of the 10 parameters analyzed with NIRs with high accuracy. The plots showed that better prediction results can be obtained since, in general, the test data samples are nearby the line adjusted for the training data. Some samples had great variability in the spectra obtained in the test dataset. However, it was considered that this variability must be reflected in the data, as it shows that the calibration models identified samples with high variability in the spectra that, therefore, should be reanalyzed in the NIR. This situation is evident in the calibrations of aW, moisture, ash, fat, protein, RT, and CT parameters. The collagen plot shows that the prediction generally results in default levels in the test group data. The parameter pigments and WHC showed that in the test subset, their levels have little variability, which can contribute to poor prediction results.

These overall results were acceptable compared to the work by Fernandez–Barroso et al. [26], with 287 Iberian pig samples of *Longissimus thoracis et lumborum*, that through PLS models obtained coefficients of determination in calibration higher than those of external validation models for both types of samples (minced and intact). In the minced samples, the parameter myoglobin showed values of R^2^_C_ of 0.84; RMSE_C_ of 0.83%; R^2^_CV_ of 0.68; RMSE_CV_ of 0.26%—except for the external validation prediction (R^2^_P_ of 0.74; and RMSE_P_ of 0.11%), whose results were worse than the present work. It is an important parameter because consumers associate the bright red color with the high meat quality of animals raised in open-air systems. For WHC, the literature data report relatively lower prediction of this parameter by using NIR spectroscopy because it is usually measured by cooking or drip loss, which affects the color and tenderness of the meat [48]. It is well known that NIR cannot directly predict cooking losses, but it can through the association of WHC with water, fat, and protein wavelengths [4]. Thus, low prediction R^2^ and RMSE in the test may be due to the small variability of the LTL muscle composition, and thus the obtained spectra, which reduced the range of calibration and effectiveness of prediction. The same conclusions were found in Wyrwisz’s work [49]. No works on pork were found in the bibliography that allow justifying the values obtained for the parameters ash, collagen, aW, RT, and CT.

## 4. Conclusions

This work shows the potential of NIR in the determination of Bísaro pork quality traits. In fact, the present work generated acceptable predictive models of meat chemical composition, using the SVMR-Poly model due to the non-linearity dependence between the spectra and the physical-chemical parameters. These models were obtained for the ash, fat, protein, collagen, and cooked texture variables. The global results can be improved considering that spectra of the same sample with great variability were used to simulate a slaughterhouse analysis environment. However, this variability can be reduced by introducing an acceptance criterion for the spectra of the same sample to obtain concordant spectra.

Nonetheless, the pigments and WHC variables had acceptable estimation models but not for prediction, mainly because the test data group presented considerable variability, in a narrow range of data. It is considered that this situation can be overcome by including new samples in the database, which could be used to increase the variability and, therefore, contribute to the robustness of the predictive models.

Overall, the present study shows that the NIR has potential as an analytical tool in situ for meat quality (nutritional) control with its inherent advantages of speed, low cost, and acceptable error, as well as not requiring a specialized technician for its handling. This way, NIR offers a promising method for classifying individual animals in breeding programs (open-air system) and applying this technique at an industrial level to obtain product characteristics of the breed.

## Figures and Tables

**Figure 1 foods-12-00470-f001:**
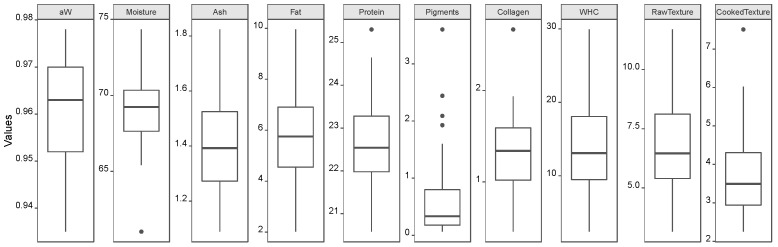
Boxplots of all the dependent variables.

**Figure 2 foods-12-00470-f002:**
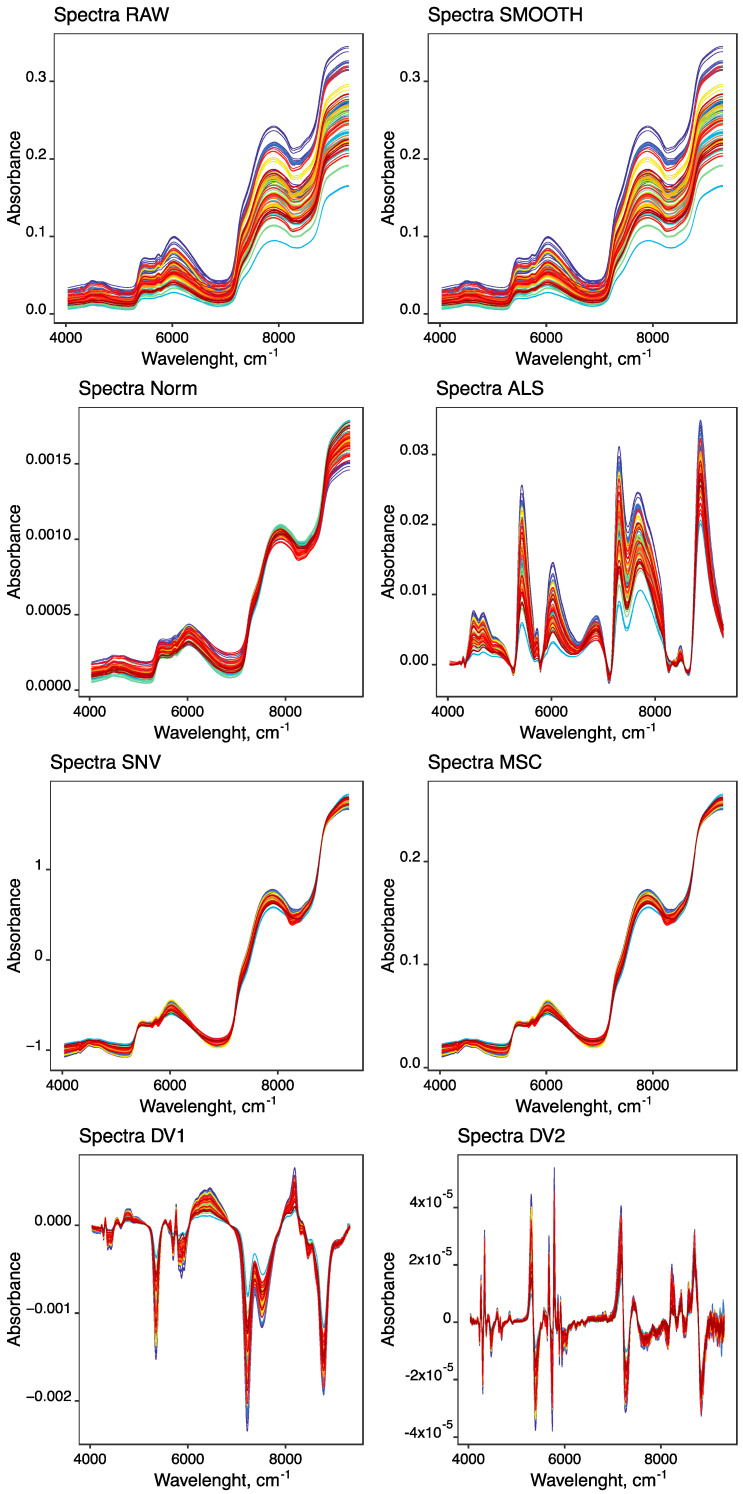
Representation of all raw, smoothed, NORM-SMT, ALS-SMT, SNV-SMT, MSC-SMT, DV1 and DV2 mean spectra obtained from the 40 samples of LTL muscles.

**Figure 3 foods-12-00470-f003:**
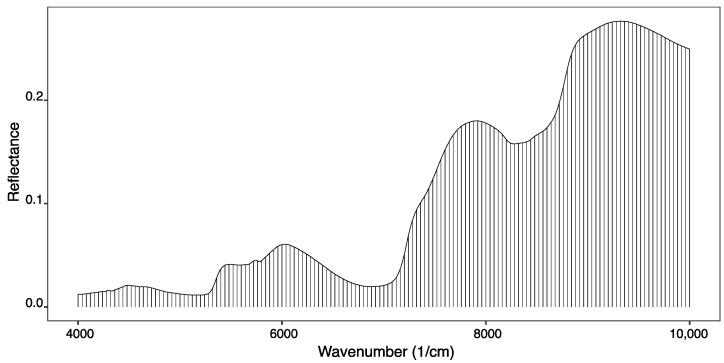
Average of the smooth spectra and the 151 points selected to represent the overall information.

**Figure 4 foods-12-00470-f004:**
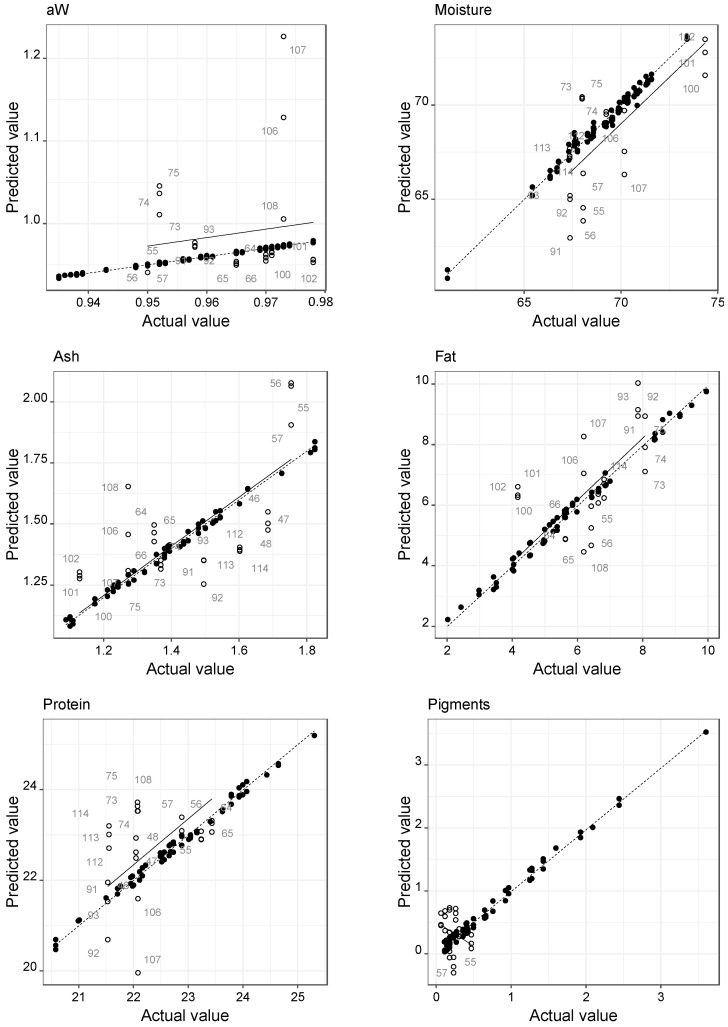
Predictive performance for all the best SVMR-Poly models obtained.

**Table 1 foods-12-00470-t001:** Dependent variables result from 40 samples of LTL muscle.

Parameter	Min	Max	Mean (±SD))
aW (%)	0.94	0.98	0.96 (±0.01)
Moisture (%)	61.04	74.34	69.03 (±2.27)
Ash (%)	1.09	1.82	1,41 (±0.19)
Collagen (%)	0.46	2.67	1.33 (±0.45)
WHC (%)	2.39	29.93	13.70 (±6.35)
Pigments	0.06	3.60	0.66 (±0.77)
CT (Kgf)	2.25	7.51	3.79 (±1.11)
RT (Kgf)	3.16	11.69	6.83 (±1.96)
Fat (%)	2.02	9.95	5.95 (±1.96)
Protein (%)	20.58	25.30	22.64 (±1.05)

SD—standard deviation; aW (%)—water activity; WHC (%)—water holding capacity; CT (%)—cooked texture; RT (%)—raw texture; pigments (mg myoglobin/g fresh muscle).

**Table 2 foods-12-00470-t002:** Average cross-validation (8 folds) results for the selected PLS and SVMR-Poly models.

Name_var	Name_df	RMSE_CV_ (±s)	MAE_CV_ (±s)	R^2^_CV_ (±s)	Best Model Parameters
PLS with CV 8 fold					
aW (%)	SMT	0.006 (±0.002)	0.005 (±0.001)	0.786 (±0.110)	20 PCs
Moisture (%)	SNV-SMT	0.839 (±0.148)	0.663 (±0.096)	0.862 (±0.099)	20 PCs
Ash (%)	NORM-SMT	0.084 (±0.021)	0.068 (±0.013)	0.829 (±0.094)	23 PCs
Fat (%)	NORM-SMT	0.520 (±0.084)	0.407 (±0.057)	0.950 (±0.017)	25 PCs
Protein (%)	NORM-SMT	0.305 (±0.066)	0.249 (±0.060)	0.924 (±0.040)	25 PCs
Pigments	SNV-SMT	0.296 (±0.055)	0.246 (±0.050)	0.861 (±0.099)	25 PCs
Collagen (%)	SNV-DV1	0.253 (±0.051)	0.208 (±0.045)	0.626 (±0.113)	14 PCs
WHC (%)	SMT	2.646 (±0.558)	2.173 (±0.396)	0.864 (±0.086)	25 PCs
RT (Kgf)	ALS-DV1	1.700 (±0.260)	1.415 (±0.253)	0.385 (±0.223)	8 PCs
CT (Kgf)	NORM-DV1	0.605 (±0.074)	0.506 (±0.065)	0.774 (±0.045)	24 PCs
SVMR-Poly with CV 8 fold					
aW (%)	MSC-SMT	0.006 (±0.003)	0.003 (±0.001)	0.825 (±0.193)	Degree = 3; Scale = 0.05; C = 1.3
Moisture (%)	SMT	0.852 (±0.312)	0.591 (±0.177)	0.863 (±0.121)	Degree = 3; Scale = 0.1; C = 1
Ash (%)	ALS-DV1	0.063 (±0.015)	0.047 (±0.012)	0.904 (±0.053)	Degree = 5; Scale = 0.007; C = 0.6
Fat (%)	ALS-DV1	0.746 (±0.142)	0.584 (±0.143)	0.883 (±0.050)	Degree = 2; Scale = 0.1; C = 0.1
Protein (%)	ALS-DV1	0.529 (±0.116)	0.417 (±0.093)	0.787 (±0.084)	Degree = 2; Scale = 0.1; C = 1
Pigments	DV1	0.229 (±0.070)	0.170 (±0.044)	0.912 (±0.087)	Degree = 2; Scale = 0.05; C = 0.5
Collagen (%)	ALS-DV1	0.202 (±0.038)	0.152 (±0.025)	0.780 (±0.070)	Degree = 5; Scale = 0.01; C = 0.5
WHC (%)	DV2	4.486 (±0.830)	3.677 (±0.660)	0.558 (±0.149)	Degree = 2; Scale = 0.1; C = 1
RT(Kgf)	DV2	1.594 (±0.232)	1.303 (±0.196)	0.447 (±0.124)	Degree = 2; Scale = 20; C = 0.5
CT (Kgf)	ALS-DV1	0.567 (±0.062)	0.433 (±0.056)	0.777 (±0.126)	Degree = 3; Scale = 0.05; C = 0.5

PLS—partial least square; SVMR-Poly–support vector machine regression Poly; SMT—smoothing; DV1—first derivative; DV2—second derivative; NORM—normalization; SNV—standard normal variate correction; MSC—multiplicative scatter correction; ALS—asymmetric least squares; PCs—principal components; RMSE_CV_—root mean square error of cross-validation; MAE_CV_—mean absolute error cross-validation; R^2^_CV_—coefficient of determination of cross-validation; WHC (%)—water holding capacity; RT (Kgf)—raw texture; CT (Kgf)—cooked texture; pigments (mg myoglobin/g fresh muscle).

**Table 3 foods-12-00470-t003:** The predictive results of the models for the training and test data.

Dependent Variable	Train Data				Train Data			
	RMSE_C_	R^2^_C_	Slope_c_	Intercept_c_	RMSE_P_	R^2^_P_	Slope_P_	Intercept_P_
PLS with CV 8 fold								
aW (%)	0.003	0.950	0.950 ± 0.022 (*p* < 0.001)	0.047 ± 0.021 (*p* = 0.029)	0.017	0.021	NS	1.401 ± 0.354 (*p* < 0.001)
Moisture (%)	0.411	0.999	0.999 ± 0.0006 (*p* < 0.001)	NS	1.811	0.009	NS	80.726 ± 11.866 (*p* < 0.001)
Ash (%)	0.029	0.999	0.999 ± 0.002 (*p* < 0.001)	NS	0.449	0.884	0.845 ± 0.021 (*p* < 0.001)	NS
Fat (%)	0.119	0.999	0.999 ± 0.020 (*p* < 0.001)	NS	3.148	0.812	1.002 ± 0.021 (*p* < 0.001)	NS
Protein (%)	0.079	0.999	0.999 ± 0.0004 (*p* < 0.001)	NS	1.404	0.996	0.953 ± 0.021 (*p* < 0.001)	NS
Pigments	0.094	0.993	0.993 ± 0.009 (*p* < 0.001)	NS	0.711	−0.029	NS	NS
Collagen (%)	0.093	0.938	0.939 ± 0.025 (*p* < 0.001)	0.081 ± 0.034 (*p* = 0.020)	0.625	0.802	0.847 ± 0.021 (*p* < 0.001)	NS
WHC (%)	0.501	0.999	0.999 ± 0.003 (*p* < 0.001)	NS	6.124	0.246	−1.041 ± 0.021 (*p* = 0.008)	22.061 ± 3.641 (*p* < 0.001)
RT (Kgf)	0.891	0.748	0.751 ± 0.044 (*p* < 0.001)	1.761 ± 0.328 (*p* < 0.001)	3.528	0.746	1.019 ± 0.120 (*p* < 0.001)	NS
CT (Kgf)	0.106	0.999	0.999 ± 0.003 (*p* < 0.001)	NS	1.784	0.829	1.137 ± 0.105 (*p* < 0.001)	NS
SVMR PF with CV 8 fold								
aW (%)	0.001	0.993	0.983 ± 0.008 (*p* < 0.001)	0.016 ± 0.008 (*p* = 0.047)	0.066	0.996	1.032 ± 0.019 (*p* < 0.001)	NS
Moisture (%)	0.294	0.982	0.973 ± 0.013 (*p* < 0.001)	1.872 ± 0.918 (*p* = 0.044)	1.972	0.999	0.986 ± 0.006 (*p* < 0.001)	NS
Ash (%)	0.016	0.992	0.967 ± 0.009 (*p* < 0.001)	0.047 ± 0.012 (*p* < 0.001)	0.192	0.983	1.006 ± 0.027 (*p* < 0.001)	NS
Fat (%)	0.180	0.992	0.965 ± 0.009 (*p* < 0.001)	0.191 ± 0.055 (*p* < 0.001)	1.289	0.965	1.007 ± 0.048 (*p* < 0.001)	NS
Protein (%)	0.089	0.993	0.952 ± 0.008 (*p* < 0.001)	1.074 ± 0.188 (*p* < 0.001)	0.940	0.998	1.015 ± 0.008 (*p* < 0.001)	NS
Pigments	0.069	0.993	0.985 ± 0.006 (*p* < 0.001)	NS	0.294	0.052	NS	0.508 ± 0.130 (*p* < 0.001)
Collagen (%)	0.032	0.994	0.955 ± 0.008 (*p* < 0.001)	0.055 ± 0.012 (*p* < 0.001)	0.248	0.947	0.702 ± 0. 034 (*p* < 0.001)	NS
WHC (%)	0.472	0.996	0.925 ± 0.006 (*p* < 0.001)	0.981 ± 0.121 (*p* < 0.001)	2.485	0.290	−0.467± 0.145 (*p* = 0.004)	16.600 ± 1.893 (*p* < 0.001)
RT (Kgf)	0.126	0.992	0.925 ± 0.006 (*p* < 0.001)	0.534 ± 0.047 (*p* < 0.001)	1.232	0.971	1.198 ± 0.042 (*p* < 0.001)	NS
CT (Kgf)	0.102	0.992	0.962 ± 0.009 (*p* < 0.001)	0.152 ± 0.036 (*p* < 0.001)	1.704	0.840	1.128 ± 0.100 (*p* < 0.001)	NS

NS—not significant (*p* > 0.05); RMSE_C_—root mean square error of calibration; R^2^_C_—coefficient of determination of calibration; RMSE_p_—root mean square error of prediction; R^2^_p_—coefficient of determination of prediction; PLS—partial least square; CV—cross-validation; SVMR PF—support vector machine regression polynomial function; WHC (%)—water holding capacity; RT (Kgf)—raw texture; CT (Kgf)—cooked texture; pigments (mg myoglobin/g fresh muscle).

## Data Availability

Data is contained within the article.

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
