# Peer review of "SVM Regression to Assess Meat Characteristics of Bísaro Pig Loins Using NIRS Methodology"

_foods, 2023, doi:10.3390/foods12030470_

Round 1

Reviewer 1 Report

It is a good work, of extreme relevance for generating information and curves for validation of the NIRS, however a question is important. The best way to validate it would be to test different samples from those used in the construction of the spectrum. Why not opt for this procedure as well?

The text presents relevant data that deserve publication.

Pay attention to the figures, as there are two figures 4, adjust to one page or divide into two figures.

Author Response

Dear Editor,

All modifications were made following the reviewer`s suggestions and comments, and responses to their comments are also attached. Thanks to their recommendations, significant modifications were made throughout the manuscript.

Thank you for your attention.

Answers to reviewer 1

Comments and Suggestions for Authors

It is a good work, of extreme relevance for generating information and curves for validation of the NIRS, however a question is important. The best way to validate it would be to test different samples from those used in the construction of the spectrum. Why not opt for this procedure as well?

We thank the reviewer for the attention given to reviewing the article and for the comments, which we took advantage of to improve the manuscript.

Yes, we agree when the reviewer says that the best procedure to validate analytical models is using different samples (test group) from those used to obtain the model (train group).

It should be noted that this is what we did in this work by dividing the samples into a train group and a test group. The reviewer's doubt shows that this point is not well clarified in the text.

We found that this information is not well explained in the summary because the estimation performance of the models is mixed with the predicting ones, and therefore external validation is not valued.

So, we made changes in the abstract in order to clarify this procedure. The phrase:

PLS models showed acceptable fits (estimation models with RMSE ≤0.5 % and R2 ≥0.95) except for RT variable (RMSE of 0.891 % and R2 of 0.748). The SVM models presented better overall prediction results than those obtained by PLS, where only the variables pigments and WHC presented estimation models (respectively: RMSE of 0.069 and 0.472 %; R2 of 0.993 and 0.996; slope of 0.985 ±0.006 and 0.925 ±0.006).”

Was changed to:

“The PLS models have estimation performance (train group data) similar to the SVM models, except for the variable RT (RMSE = 0.89% and R2 = 0.75). SVM models showed better overall prediction results (test group data) than those obtained by PLS. In general, SVM models had acceptable prediction results (RMSE < 2.5 and R2 > 0.94) for the variables aW, moisture, ash, fat, protein, collagen and raw texture. Poor predicting performance was obtained for the cooked texture variable (RMSE = 1.7 and R2 = 0.84) while, the pigments and WHC variables only presented estimation models.”

Also, In the caption of Figure 4 and 5, the distinction between the two markers was introduced: bold marker – train group; delimited marker - test group.

The text presents relevant data that deserve publication.

Pay attention to the figures, as there are two figures 4, adjust to one page or divide into two figures.

Changes have been made in the revised version of the manuscript according to the suggestion.

Reviewer 2 Report

This article SVM regression to assess meat characteristics of Bísaro pig loins using NIRS methodology investigated an interesting  issue of porcine muscle. Developing a rapid and non-destructive method for food safety, quality and authenticity monitoring has become a crucial request from the meat industry. The manuscript is concise, well-written, and scientifically sound. The language of the manuscript needs revision as there are many areas for improvement and corrections. The bibliography also required major revision. I have comments on specifics as follows.
L 3: Write “methodology” instead of “methology”
or any suitbale term regarding mathematics.

L 18:  Add “s” with estimate.

L 20: Write “an” instead of “a” after scanned in.

L 23: Write “except” instead of “with the exception of”.

L 27: Write “predict” instead of “prediction”

L 28: “brand” à In your whole abstract, you are using short forms irregularly, if you have word limit problem then you can reduce some content in another way. In case of famous terminologies it's fine but try to avoid uncommon terms.

L 38: “has” after It also à  The phrase has the ability to may be unnecessarily wordy. Consider replacing the phrase with a simpler alternative.

L 38:  Write “can” instead of “the ability to”.

L 40: Write “cost-effective” instead of “cost effective”.

L 42:  “Norms”à Add reference here

L 47: Delete “,” after characterize foods.

L 48: “processes” à Please add citation.

L 48: Delete “arising” after bands.

L 56: Add “,” after pre-treatment techniques.

L 58: “DV1” à Initially use full form.

L 60: Add “such” before as principal.

L 62: Add “the” before qualitative.

L 68:  Write “for” instead of “to” before modelling purposes.

L 68:  Write “modeling” instead of “modelling”.

L 75: Delete “,” after and àwrite “and springiness”.

L 78: Add “and” before pH of commercial.

L 82:  Write “The meat industry” instead of “Meat industry”.

L 86:  Write “meat-related” instead of “meat related”.

L 88: Add “s” with origin after graphical.

L 94: “(PDO)”  à ???

L 111:  Add “the” before value of animals.

L 114:  group à How many?

L 121:  Delete “,” after then.

L 127:  Delete “,” after (Portugal).

L 136:  Add “the” before sample.

L 145:  Write “weighed” instead of “weighted”.

L 186:  Add “the” before correlation.

L 188:  Write “were” instead of “was” before divided.

L 194:  Delete “an” before external.

L 197:  Write “were” instead of “was” before applied.

L 199:  Write “of” instead of “a” before limited.

L 202:  Delete “,” after analysis.

L 207:  Add “,” after results.

L 207:  Check punctuation.

L 212: Delete “for” before estimation.

L 232:  Write “variables result” instead of “variables results”.

L 248:  Delete “,” after samples.

L 259:  spectral à It appears that the modifiers in the noun phrase calibration spectral information are in the wrong order. Consider changing the word order.

L 268:  “All results above mentioned agree” àThere seems to be a word order problem here.

L 275:  Delete “the” before Figure 1.

L 278:  Delete “any” before variable.

L 279:  Add “a” before normal.

L 291:  Write “was” instead of “were” before used.

L 304:  Add “it” after since.

L 310:  Delete “the” before representative.

L 317:  Write “were” instead of “was” before divided.

L 346:  Write “shown” instead of “showed”.

L 350:  “non-linear” à Please use same pattern throughout the draft. Somewhere – used somewhere didn’t.

L 359:  Delete “,” after obtained.

L 359: Add “,” after since.

L 359: Add “,” after test data.

L 381:  Add “,” after test data.

L 403:  Write “Some samples had great variability in the spectra obtained in the test data set” instead of “In the test data set, some samples had great variability in the spectra obtained”.

L 409: Add “,” after subset.

L 420: Delete “,” after spectroscopy.

L 420: Add “it” after because.

L 425: Delete “some”

L 425: “conclusions” à The same

L 426: Write “in” instead of “by” after found.

L 470: Delete “the” after using.

L 475: Delete “,” after sample.

L 480: Add “were” after database.

L 505: “References” à Cross check literature cited.

L 522: “7. Osborne, B.G. Near‐infrared spectroscopy in food analysis. Encyclopedia of analytical chemistry: applications, theory and instrumentation. 2006. https://doi.org/10.1002/9780470027318.a1018” à Recheck

L 558: “2019” à Font Bold

L 578: “23–25” à Please check journal reference format. i.e

L 613: “46. Varmuza, K.; Filzmoser, P. Introduction to multivariate statistical analysis in chemometrics. CRC Press, 2009.” à Pages??

Author Response

Dear Editor,

All modifications were made following the reviewer`s, suggestions and comments, and

responses to their comments are also attached. Thanks to their recommendations,

significant modifications were made throughout the manuscript.

Thank you for your attention.

Answers to reviewer 2

Comments and Suggestions for Authors

This article SVM regression to assess meat characteristics of Bísaro pig loins using NIRS methodology investigated an interesting  issue of porcine muscle. Developing a rapid and non-destructive method for food safety, quality and authenticity monitoring has become a crucial request from the meat industry. The manuscript is concise, well-written, and scientifically sound. The language of the manuscript needs revision as there are many areas for improvement and corrections. The bibliography also required major revision. I have comments on specifics as follows.

Response: We thank the reviewer for the attention given to reviewing the article and for the comments, which we took advantage of to improve the manuscript.

The manuscript was revised by an native English citizen.

L 3: Write “methodology” instead of “methology” or any suitable term regarding mathematics.

Response: Suggestion accepted; changes have been made in the revised version of the

manuscript.

L 18:  Add “s” with estimate.

Response: Suggestion accepted; changes have been made in the revised version of the

manuscript.

L 20: Write “an” instead of “a” after scanned in.

Response: Suggestion accepted; changes have been made in the revised version of the

manuscript.

L 23: Write “except” instead of “with the exception of”.

Response: Suggestion accepted; changes have been made in the revised version of the

manuscript.

L 27: Write “predict” instead of “prediction”

Response: Suggestion accepted; changes have been made in the revised version of the

manuscript.

L 28: “brand” à In your whole abstract, you are using short forms irregularly, if you have word limit problem then you can reduce some content in another way. In case of famous terminologies it's fine but try to avoid uncommon terms.

Response:  We appreciate the suggestion; changes have been made in the revised version of the manuscript.

L 38: “has” after It also à  The phrase has the ability to may be unnecessarily wordy. Consider replacing the phrase with a simpler alternative.

Response: Suggestion accepted; changes have been made in the revised version of the

manuscript.

L 38:  Write “can” instead of “the ability to”.

Response: Suggestion accepted; changes have been made in the revised version of the

manuscript.

L 40: Write “cost-effective” instead of “cost effective”.

Response: Suggestion accepted; changes have been made in the revised version of the

manuscript.

L 42:  “Norms”à Add reference here

Response: We chose to remove this part of the text because it would not make sense to mention the references and norms here, since they are mentioned later

L 47: Delete “,” after characterize foods.

Response: Suggestion accepted; changes have been made in the revised version of the

manuscript.

L 48: “processes” à Please add citation.

Response: Suggestion accepted; changes have been made in the revised version of the

manuscript.

L 48: Delete “arising” after bands.

Response: Suggestion accepted; changes have been made in the revised version of the

manuscript.

L 56: Add “,” after pre-treatment techniques.

Response: Suggestion accepted; changes have been made in the revised version of the

manuscript.

L 58: “DV1” à Initially use full form.

Response: Suggestion accepted; changes have been made in the revised version of the

manuscript.

L 60: Add “such” before as principal.

Response: Suggestion accepted; changes have been made in the revised version of the

manuscript.

L 62: Add “the” before qualitative.

Response: Suggestion accepted; changes have been made in the revised version of the

manuscript.

L 68:  Write “for” instead of “to” before modelling purposes.

Response: Suggestion accepted; changes have been made in the revised version of the

manuscript.

L 68:  Write “modeling” instead of “modelling”.

Response: Suggestion accepted; changes have been made in the revised version of the

manuscript.

L 75: Delete “,” after and àwrite “and springiness”.

Response: Suggestion accepted; changes have been made in the revised version of the

manuscript.

L 78: Add “and” before pH of commercial.

Response: Suggestion accepted; changes have been made in the revised version of the

manuscript.

L 82:  Write “The meat industry” instead of “Meat industry”.

Response: Suggestion accepted; changes have been made in the revised version of the

manuscript.

L 86:  Write “meat-related” instead of “meat related”.

Response: Suggestion accepted; changes have been made in the revised version of the

manuscript.

L 88: Add “s” with origin after graphical.

Response: Suggestion accepted; changes have been made in the revised version of the

manuscript.

L 94: “(PDO)”  à ???

Response: Suggestion accepted; changes have been made in the revised version of the

manuscript. The word "origin" corresponding to the nomenclature, which has already been corrected, was missing.

L 111:  Add “the” before value of animals.

Response: Suggestion accepted; changes have been made in the revised version of the

manuscript.

L 114:  group à How many?

Response: Suggestion accepted; changes have been made in the revised version of the

manuscript.

L 121:  Delete “,” after then.

Response: Suggestion accepted; changes have been made in the revised version of the

manuscript.

L 127:  Delete “,” after (Portugal).

Response: Suggestion accepted; changes have been made in the revised version of the

manuscript.

L 136:  Add “the” before sample.

Response: Suggestion accepted; changes have been made in the revised version of the

manuscript.

L 145:  Write “weighed” instead of “weighted”.

Response: Suggestion accepted; changes have been made in the revised version of the

manuscript.

L 186:  Add “the” before correlation.

Response: Suggestion accepted; changes have been made in the revised version of the

manuscript.

L 188:  Write “were” instead of “was” before divided.

Response: Suggestion accepted; changes have been made in the revised version of the

manuscript.

L 194:  Delete “an” before external.

Response: Suggestion accepted; changes have been made in the revised version of the

manuscript.

L 197:  Write “were” instead of “was” before applied.

Response: Suggestion accepted; changes have been made in the revised version of the

manuscript.

L 199:  Write “of” instead of “a” before limited.

Response: Suggestion accepted; changes have been made in the revised version of the

manuscript.

L 202:  Delete “,” after analysis.

Response: Suggestion accepted; changes have been made in the revised version of the

manuscript.

L 207:  Add “,” after results.

Response: Suggestion accepted; changes have been made in the revised version of the

manuscript.

L 207:  Check punctuation.

Response: Suggestion accepted; changes have been made in the revised version of the

manuscript.

L 212: Delete “for” before estimation.

Response: Suggestion accepted; changes have been made in the revised version of the

manuscript.

L 232:  Write “variables result” instead of “variables results”.

Response: Suggestion accepted; changes have been made in the revised version of the

manuscript.

L 248:  Delete “,” after samples.

Response: Suggestion accepted; changes have been made in the revised version of the

manuscript.

L 259:  spectral à It appears that the modifiers in the noun phrase calibration spectral information are in the wrong order. Consider changing the word order.

Response: Suggestion accepted; changes have been made in the revised version of the

manuscript.

L 268: “All results above mentioned agree” àThere seems to be a word order problem here.

Response: Suggestion accepted; changes have been made in the revised version of the

manuscript.

L 275:  Delete “the” before Figure 1.

Response: Suggestion accepted; changes have been made in the revised version of the

manuscript.

L 278:  Delete “any” before variable.

Response: Suggestion accepted; changes have been made in the revised version of the

manuscript.

L 279:  Add “a” before normal.

Response: Suggestion accepted; changes have been made in the revised version of the

manuscript.

L 291:  Write “was” instead of “were” before used.

Response: Suggestion accepted; changes have been made in the revised version of the

manuscript.

L 304:  Add “it” after since.

Response: Suggestion accepted; changes have been made in the revised version of the

manuscript.

L 310:  Delete “the” before representative.

Response: Suggestion accepted; changes have been made in the revised version of the

manuscript.

L 317:  Write “were” instead of “was” before divided.

Response: Suggestion accepted; changes have been made in the revised version of the

manuscript.

L 346:  Write “shown” instead of “showed”.

Response: Suggestion accepted; changes have been made in the revised version of the

manuscript.

L 350:  “non-linear” à Please use same pattern throughout the draft. Somewhere – used somewhere didn’t.

Response: Suggestion accepted; changes have been made in the revised version of the

manuscript.

L 359:  Delete “,” after obtained.

Response: Suggestion accepted; changes have been made in the revised version of the

manuscript.

L 359: Add “,” after since.

Response: Suggestion accepted; changes have been made in the revised version of the

manuscript.

L 359: Add “,” after test data.

Response: Suggestion accepted; changes have been made in the revised version of the

manuscript.

L 381:  Add “,” after test data.

Response: Suggestion accepted; changes have been made in the revised version of the

manuscript.

L 403:  Write “Some samples had great variability in the spectra obtained in the test data set” instead of “In the test data set, some samples had great variability in the spectra obtained”.

Response: Suggestion accepted; changes have been made in the revised version of the

manuscript.

L 409: Add “,” after subset.

Response: Suggestion accepted; changes have been made in the revised version of the

manuscript.

L 420: Delete “,” after spectroscopy.

Response: Suggestion accepted; changes have been made in the revised version of the

manuscript.

L 420: Add “it” after because.

Response: Suggestion accepted; changes have been made in the revised version of the

manuscript.

L 425: Delete “some”

Response: Suggestion accepted; changes have been made in the revised version of the

manuscript.

L 425: “conclusions” à The same

Response: Suggestion accepted; changes have been made in the revised version of the

manuscript.

L 426: Write “in” instead of “by” after found.

Response: Suggestion accepted; changes have been made in the revised version of the

manuscript.

L 470: Delete “the” after using.

Response: Suggestion accepted; changes have been made in the revised version of the

manuscript.

L 475: Delete “,” after sample.

Response: Suggestion accepted; changes have been made in the revised version of the

manuscript.

L 480: Add “were” after database.

Response: Suggestion accepted; changes have been made in the revised version of the

manuscript.

L 505: “References” à Cross check literature cited.

Response: we don't understand what you want in your suggestion. can you rephrase it?

L 522: “7. Osborne, B.G. Near‐infrared spectroscopy in food analysis. Encyclopedia of analytical chemistry: applications, theory and instrumentation. 2006. https://doi.org/10.1002/9780470027318.a1018” à Recheck

Response: After reviewing the reference, we realize that the same DOI, refers to online links of different textual parts belonging to the same encyclopedia.  The article consulted refers to the following information  “Encyclopedia of Analytical Chemistry, Online © 2006 John Wiley & Sons, Ltd. This article is © 2006 John Wiley & Sons, Ltd. This article was published in the Encyclopedia of Analytical Chemistry in 2006 by John Wiley & Sons, Ltd. DOI: 10.1002/9780470027318.a1018”

L 558: “2019” à Font Bold

Response: Suggestion accepted; changes have been made in the revised version of the

Manuscript.

L 578: “23–25” à Please check journal reference format. i.e

Response: Suggestion accepted; changes have been made in the revised version of the

Manuscript.

L 613: “46. Varmuza, K.; Filzmoser, P. Introduction to multivariate statistical analysis in chemometrics. CRC Press, 2009.” à Pages??

Response: Suggestion accepted; changes have been made in the revised version of the

Manuscript.
